# VIEWMAKER NETWORKS: LEARNING VIEWS FOR UNSUPERVISED REPRESENTATION LEARNING

**Alex Tamkin, Mike Wu, Noah Goodman**
Department of Computer Science
Stanford University
Stanford, CA 94305, USA
`{atamkin, wumike, ngoodman}@stanford.edu`

## ABSTRACT

Many recent methods for unsupervised representation learning train models to be invariant to different "views," or distorted versions of an input. However, designing these views requires considerable trial and error by human experts, hindering widespread adoption of unsupervised representation learning methods across domains and modalities. To address this, we propose *viewmaker networks*: generative models that learn to produce useful views from a given input. Viewmakers are *stochastic bounded adversaries*: they produce views by generating and then adding an $\ell_p$-bounded perturbation to the input, and are trained adversarially with respect to the main encoder network. Remarkably, when pretraining on CIFAR-10, our learned views enable comparable transfer accuracy to the well-tuned SimCLR augmentations—despite not including transformations like cropping or color jitter. Furthermore, our learned views significantly outperform baseline augmentations on speech recordings (+9 points on average) and wearable sensor data (+17 points on average). Viewmaker views can also be combined with handcrafted views: they improve robustness to common image corruptions and can increase transfer performance in cases where handcrafted views are less explored. These results suggest that viewmakers may provide a path towards more general representation learning algorithms—reducing the domain expertise and effort needed to pretrain on a much wider set of domains. Code is available at `https://github.com/alextamkin/viewmaker`.

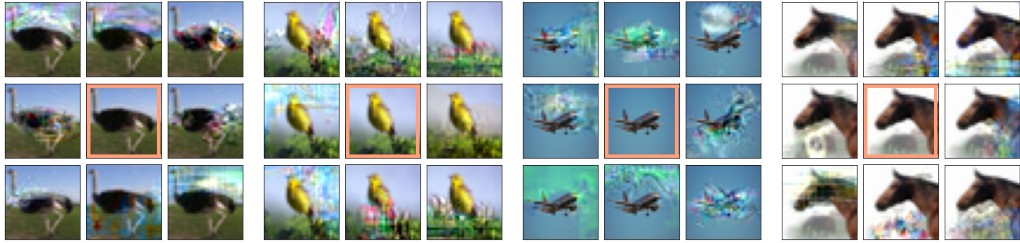

Figure 1: **Viewmaker networks generate complex and diverse input-dependent views for unsupervised learning.** Examples shown are for CIFAR-10. Original image in center with pink border.

## 1 INTRODUCTION

Unsupervised representation learning has made significant recent strides, including in computer vision, where view-based methods have enabled strong performance on benchmark tasks (Wu et al., 2018; Oord et al., 2018; Bachman et al., 2019; Zhuang et al., 2019; Misra & Maaten, 2020; He et al., 2020; Chen et al., 2020a). *Views* here refer to human-defined data transformations, which target capabilities or invariances thought to be useful for transfer tasks. In particular, in contrastive learning of visual representations, models are trained to maximize the mutual information between different views of an image, including crops, blurs, noise, and changes to color and contrast (Bachman et al.,

2019; Chen et al., 2020a). Much work has investigated the space of possible image views (and their compositions) and understanding their effects on transfer learning (Chen et al., 2020a; Wu et al., 2020; Tian et al., 2019; Purushwalkam & Gupta, 2020).

The fact that views must be hand designed is a significant limitation. While views for image classification have been refined over many years, new views must be developed from scratch for new modalities. Making matters worse, even *within* a modality, different domains may have different optimal views (Purushwalkam & Gupta, 2020). Previous studies have investigated the properties of good views through the lens of mutual information (Tian et al., 2020; Wu et al., 2020), but a broadly-applicable approach for learning views remains unstudied.

In this work, we present a general method for learning diverse and useful views for contrastive learning. Rather than searching through possible compositions of existing view functions (Cubuk et al., 2018; Lim et al., 2019), which may not be available for many modalities, our approach produces views with a generative model, called the *viewmaker* network, trained jointly with the encoder network. This flexibility enables learning a broad set of possible view functions, including input-dependent views, without resorting to hand-crafting or expert domain knowledge. The viewmaker network is trained adversarially to create views which increase the contrastive loss of the encoder network. Rather than directly outputting views for an image, the viewmaker instead outputs a stochastic perturbation that is *added* to the input. This perturbation is projected onto an $\ell_p$ sphere, controlling the effective strength of the view, similar to methods in adversarial robustness. This constrained adversarial training method enables the model to reduce the mutual information between different views while preserving useful input features for the encoder to learn from.

In summary, we contribute:

1. Viewmaker networks: to our knowledge the first modality-agnostic method to *learn* views for unsupervised representation learning
2. On image data, where expert-designed views have been extensively optimized, our viewmaker-models achieve comparable transfer performance to state of the art contrastive methods while being more robust to common corruptions.
3. On speech data, our method significantly outperforms existing human-defined views on a range of speech recognition transfer tasks.
4. On time-series data from wearable sensors, our model significantly outperforms baseline views on the task of human activity recognition (e.g., cycling, running, jumping rope).

## 2 RELATED WORK

**Unsupervised representation learning** Learning useful representations from unlabeled data is a fundamental problem in machine learning (Pan & Yang, 2009; Bengio et al., 2013). A recently successful framework for unsupervised representation learning for images involves training a model to be invariant to various data transformations (Bachman et al., 2019; Misra & Maaten, 2020), although the idea has much earlier roots (Becker & Hinton, 1992; Hadsell et al., 2006; Dosovitskiy et al., 2014). This idea has been expanded by a number of contrastive learning approaches which push embeddings of different views, or transformed inputs, closer together, while pushing other pairs apart (Tian et al., 2019; He et al., 2020; Chen et al., 2020a;b;c), as well as non-contrastive approaches which do not explicitly push apart unmatched views (Grill et al., 2020; Caron et al., 2020). Related but more limited setups have been explored for speech, where data augmentation strategies are less explored (Oord et al., 2018; Kharitonov et al., 2020).

**Understanding and designing views** Several works have studied the role of views in contrastive learning, including from a mutual-information perspective (Wu et al., 2020), in relation to specific transfer tasks (Tian et al., 2019), with respect to different kinds of invariances (Purushwalkam & Gupta, 2020), or via careful empirical studies (Chen et al., 2020a). Outside of a contrastive learning framework, Gontijo-Lopes et al. (2020) study how data augmentation aids generalization in vision models. Much work has explored different handcrafted data augmentation methods for supervised learning of images (Hendrycks et al., 2020; Lopes et al., 2019; Perez & Wang, 2017; Yun et al., 2019; Zhang et al., 2017), speech (Park et al., 2019; Kovács et al., 2017; Tóth et al., 2018; Kharitonov et al., 2020), or in feature space (DeVries & Taylor, 2017).

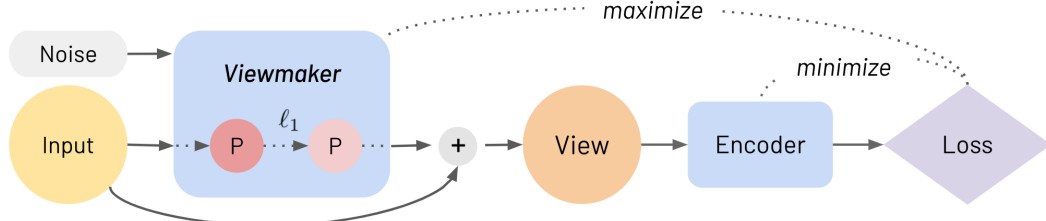

Figure 2: **Diagram of our method.** The viewmaker network is trained to produce stochastic adversarial views restricted to an $\ell_1$ sphere around the input.

**Adversarial methods**   Our work is related to and inspired by work on adversarial methods, including the $\ell_p$ balls studied in adversarial robustness (Szegedy et al., 2013; Madry et al., 2017; Raghunathan et al., 2018) and training networks with adversarial objectives (Goodfellow et al., 2014; Xiao et al., 2018). Our work is also connected to the vicinal risk minimization principle (Chapelle et al., 2001) and can be interpreted as producing amortized virtual adversarial examples (Miyato et al., 2018). Previous adversarial view-based pretraining methods add adversarial noise on top of existing handcrafted views (Kim et al., 2020) or require access to specific transfer tasks during pretraining (Tian et al., 2020). In contrast, our method is more general: it is neither specialized to a particular downstream task, nor requires neither human-defined view families. Outside of multi-view learning paradigms, adversarial methods have also seen use for representation learning in GANs (Donahue et al., 2016; Donahue & Simonyan, 2019) or in choosing harder negative samples (Bose et al., 2018), as well as for data augmentation (Antoniou et al., 2017; Volpi et al., 2018; Bowles et al., 2018). Adversarial networks that perturb inputs have also been investigated to improve GAN training (Sajjadi et al., 2018) and to remove "shortcut" features (e.g., watermarks) for self-supervised pretext tasks (Minderer et al., 2020).

**Learning views**   Outside of adversarial approaches, our work is related to other studies that seek to learn data augmentation strategies by composing existing human-designed augmentations (Ratner et al., 2017; Cubuk et al., 2018; Zhang et al., 2019; Ho et al., 2019; Lim et al., 2019; Cubuk et al., 2020) or by modeling variations specific to the data distribution (Tran et al., 2017; Wong & Kolter, 2020). By contrast, our method requires no human-defined view functions, does not require first pretraining a generative model, and can generate perturbations beyond naturally-occurring variation observed in the training data (e.g. brightness or contrast), potentially conferring robustness benefits, as we explore in Section 4.3.

## 3   METHOD

In contrastive learning, the objective is to push embeddings of positive views (derived from the same input) close together, while pushing away embeddings of negative views (derived from different inputs). We focus mainly on the simple, yet performant, SimCLR contrastive learning algorithm (Chen et al., 2020a), but we also consider a memory bank-based algorithm (Wu et al., 2018) in Section 4. As our method is agnostic to the specific pretraining loss used, it is naturally compatible with other view-based algorithms such as MoCo (He et al., 2020), BYOL (Grill et al., 2020), and SwAV (Caron et al., 2020) by similarly substituting the data transformation pipeline with a viewmaker network.

Formally, given a batch of $N$ pairs of positive views $(i, j)$ the SimCLR loss is

$$\mathcal{L} = \frac{1}{2N} \sum_{k=1}^{N} [\ell(2k-1, 2k) + \ell(2k, 2k-1)] \text{ where } \ell(i, j) = -\log \frac{\exp(s_{i,j}/\tau)}{\sum_{k=1}^{2N} \mathbb{1}_{[k \neq i]} \exp(s_{i,k}/\tau)}$$

and $s_{a,b}$ is the cosine similarity of the embeddings of views $a$ and $b$.

We generate views by perturbing examples with a *viewmaker* network $V$, trained jointly with the main *encoder* network $M$. There are three attributes desirable for useful perturbations, each of which motivates an aspect of our method:

1. **Challenging:** The perturbations should be complex and strong enough that an encoder must develop useful representations to perform the self-supervised task. We accomplish this by generating perturbations with a neural network that is trained adversarially to increase the loss of the encoder network. Specifically, we use a neural network that ingests the input $X$ and outputs a view $X + V(X)$.

2. **Faithful:** The perturbations must not make the encoder task impossible, being so strong that they destroy all features of the input. For example, perturbations should not be able to zero out the input, making learning impossible. We accomplish this by constraining the perturbations to an $\ell_p$ sphere around the original input. $\ell_p$ constraints are common in the adversarial robustness literature where perturbations are expected to be indistinguishable. In our experiments, we find the best results are achieved with an $\ell_1$ sphere, which grants the viewmaker a *distortion budget* that it can spend on a small perturbation for a large part of the input or a more extreme perturbation for a smaller portion.

3. **Stochastic:** The method should be able to generate a variety of perturbations for a single input, as the encoder objective requires contrasting two different views of an input against each other. To do this, we inject random noise into the viewmaker, such that the model can learn a stochastic function that produces a different perturbed input each forward pass.

Figure 2 summarizes our method. The encoder and viewmaker are optimized in alternating steps to minimize and maximize $\mathcal{L}$, respectively. We use an image-to-image neural network as our viewmaker network, with an architecture adapted from work on style transfer (Johnson et al., 2016). See the Appendix for more details. This network ingests the input image and outputs a perturbation that is constrained to an $\ell_1$ sphere. The sphere's radius is determined by the volume of the input tensor times a hyperparameter $\epsilon$, the *distortion budget*, which determines the strength of the applied perturbation. This perturbation is added to the input image and optionally clamped in the case of images to ensure all pixels are in $[0, 1]$. Algorithm 1 describes this process precisely.

---

**Algorithm 1:** Generating viewmaker views

---

**Input:** Viewmaker network $V$, $C \times W \times H$ image X, $\ell_1$ distortion budget $\epsilon$, noise $\delta$
**Output:** Perturbed $C \times W \times H$ image $X$
$P \leftarrow V(X, \delta)$ // generate perturbation
$P \leftarrow \frac{\epsilon CWH}{|P|_1} P$ // project to $\ell_1$ sphere
$X \leftarrow X + P$ // apply perturbation
$X \leftarrow \text{clamp}(X, 0, 1)$ // clamp (images only)

---

## 4  IMAGES

We begin by applying the viewmaker to contrastive learning for images. In addition to SimCLR (Chen et al., 2020a), we also consider a memory bank-based instance discrimination framework (Wu et al., 2018, henceforth InstDisc).

We pretrain ResNet-18 (He et al., 2015) models on CIFAR-10 (Krizhevsky, 2009) for 200 epochs with a batch size of 256. We train a viewmaker-encoder system with a distortion budget of $\epsilon = 0.05$. We tried distortion budgets $\epsilon \in \{0.1, 0.05, 0.02\}$ and found $0.05$ to work best; however, we anticipate that further tuning would yield additional gains. As we can see in Figure 1, the learned views are diverse, consisting of qualitatively different kinds of perturbations and affecting different parts of the input. We compare the resulting encoder representations with a model trained with the *expert views* used for SimCLR, comprised of many human-defined transformations targeting different kinds of invariances useful for image classification: cropping-and-resizing, blurring, horizontal flipping, color dropping, and shifts in brightness, contrast, saturation, and hue (Chen et al., 2020a).

### 4.1  TRANSFER RESULTS ON IMAGE CLASSIFICATION TASKS

We evaluate our models on CIFAR-10, as well as eleven transfer tasks including MetaDataset (Triantafillou et al., 2019), MSCOCO (Lin et al., 2014), MNIST (LeCun et al., 1998), and FashionMNIST (Xiao et al., 2017). We use the standard linear evaluation protocol, which trains a logistic

| Dataset | SimCLR | | InstDisc | | Dataset | SimCLR | | InstDisc | |
|---|---|---|---|---|---|---|---|---|---|
| | Expt | Ours | Expt | Ours | | Expt | Ours | Expt | Ours |
| CIFAR-10 | **86.2** | 84.5 | **82.4** | 80.1 | MNIST | 97.1 | **98.7** | 98.7 | **98.9** |
| MSCOCO | 49.9 | **50.4** | 48.6 | **50.2** | FaMNIST | 88.3 | **91.5** | 89.2 | **91.4** |
| CelebA (F1) | 51.0 | **51.8** | **57.0** | 53.7 | CUBirds | **11.2** | 8.7 | **13.7** | 9.4 |
| LSUN | **56.2** | 55.0 | **56.0** | 55.6 | VGGFlower | 53.3 | **53.6** | **61.5** | 54.8 |
| Aircraft | **32.5** | 31.7 | **37.7** | 33.5 | TrafficSign | **96.6** | 94.9 | **98.9** | 94.3 |
| DTD | **30.4** | 28.8 | **29.8** | 29.8 | Fungi | **2.2** | 2.0 | **2.6** | 2.1 |

Table 1: **Our learned views (Ours) enable comparable transfer performance to expert views (Expt) on CIFAR-10.** Suite of transfer tasks using pretrained representations from CIFAR-10 for both the SimCLR and InstDisc pretraining setups. Numbers are percent accuracy with the exception of CelebA which is F1. FaMNIST stands for FashionMNIST.

regression on top of representations from a frozen model. We apply the same views as in pretraining, freezing the final viewmaker when using learned views; we apply no views during validation. Table 1 shows our results, indicating comparable overall performance with SimCLR and InstDisc, all without the use of human-crafted view functions. This performance is noteworthy as our $\ell_1$ views cannot implement cropping-and-rescaling, which was shown to be the most important view function in Chen et al. (2020a). We speculate that the ability of the viewmaker to implement partial masking of an image may enable a similar kind of spatial information ablation as cropping.

### 4.1.1 COMPARISON TO RANDOM $\ell_1$ NOISE

Is random noise sufficient to produce domain-agnostic views? To assess how important adversarial training is to the quality of the learned representations, we perform an ablation where we generate views by adding Gaussian noise normalized to the same $\epsilon = 0.05$ budget as used in the previous section. Transfer accuracy on CIFAR-10 is significantly hurt by this ablation, reaching **52.01%** for a SimCLR model trained with random noise views compared to **84.50%** for our method, demonstrating the importance of adversarial training to our method.

### 4.1.2 THE IMPORTANCE OF INTER-PATCH MUTUAL INFORMATION AND CROPPING VIEWS

Cropping-and-resizing has been identified as a crucial view function when pretraining on ImageNet (Chen et al., 2020a). However, what properties of a pretraining dataset make cropping useful? We hypothesize that such a dataset must have images whose patches have high mutual information. In other words, there must be some way for the model to identify that different patches of the same image come from the same image. While this may be true for many object or scene recognition datasets, it may be false for other important pretraining datasets, including medical or satellite imagery, where features of interest are isolated to particular parts of the image.

To investigate this hypothesis, we modify the CIFAR-10 dataset to reduce the inter-patch mutual information by replacing each 16x16 corner of the image with the corner from another image in the training dataset (see Figure 3 for an example). Thus, random crops on this dataset, which we call CIFAR-10-Corners, will often contain completely unrelated information. When pretrained on CIFAR-10-Corners, expert views achieve **63.3%** linear evaluation accuracy on the original CIFAR-10 dataset, while viewmaker views achieve **68.8%**. This gap suggests that viewmaker views are less reliant on inter-patch mutual information than the expert views.

### 4.2 COMBINING VIEWMAKER AND HANDCRAFTED VIEWS

Can viewmakers improve performance in cases where some useful handcrafted views have already been identified? Chen et al. (2020a) show that views produced through cropping are significantly improved by a suite of color-based augmentations, which they argue prevents the network from relying solely on color statistics to perform the contrastive task. Here, we show that viewmaker networks also enable strong gains when added on top of cropping and horizontal flipping views when pretraining on CIFAR-10—without any domain-specific knowledge. Alone, this subset of

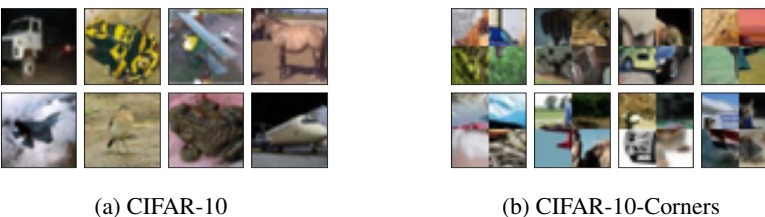

|                    |                     |
|--------------------|---------------------|
| (a) CIFAR-10       | (b) CIFAR-10-Corners |

Figure 3: Our learned views are still able to yield useful information even when the inter-patch mutual information in a dataset is low, as in Figure 3b.

| Views     | Clean    | Corrupted | Diff  |
|-----------|----------|-----------|-------|
| Ours      | 84.5     | 71.4      | -13.1 |
| SimCLR*   | **86.2** | 77.1      | -9.1  |
| Combined* | **86.3** | **79.8**  | **-6.5** |

(a) Accuracy on CIFAR-10 and CIFAR-10-C.
*Overlap with CIFAR-10-C corruptions.

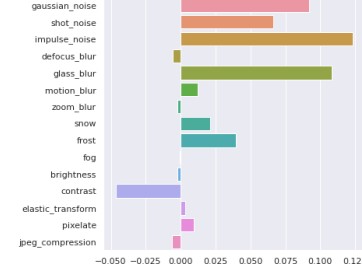

(b) Accuracy gain on CIFAR-10-C by from adding our learned views atop expert views.

Figure 4: **Performance of different views on CIFAR-10-C corruptions.** Our learned views enable solid performance in the face of unseen corruptions despite not explicitly including any blurring, contrast, or brightness transformations during training, unlike the expert views. Adding our learned views on top of SimCLR yields additional gains in robust accuracy, especially on different kinds of noise corruptions and glass blurring.

handcrafted augmentations achieves **73.2%** linear evaluation accuracy on CIFAR-10. Combining these views with learned viewmaker perturbations ($\epsilon = 0.05$) achieves **83.1%**.[1] This suggests that viewmakers can significantly improve representation learning even in cases where some domain-specific views have already been developed.

### 4.3 ROBUSTNESS TO COMMON CORRUPTIONS

Image classification systems should behave robustly even when the data distribution is slightly different from that seen during training. Does using a viewmaker improve robustness against common types of corruptions not experienced at train time? To answer this, we evaluate both learned views, expert views, and their composition on the CIFAR-10-C dataset (Hendrycks & Dietterich, 2019), which assesses robustness to corruptions like snow, pixelation, and blurring. In this setting, corruptions are applied only at test time, evaluating whether the classification system is robust to some types of corruptions to which humans are robust.

When considering methods in isolation, SimCLR augmentations result in less of an accuracy drop from clean to corrupted data compared to our learned views, as shown in Table 4a. This gap is expected, as the expert views overlap significantly with the CIFAR-10-C corruptions: both include blurring, brightness, and contrast transformations. Interestingly, however, when we train a viewmaker network while also applying expert augmentations ("Combined," Table 4a), we can further improve the robust accuracy, with notable gains on noise and glass blur corruptions (Figure 4b). This is noteworthy, as our learned views have no explicit overlap with the CIFAR-10-C corruptions, unlike the expert augmentations.[2] In the Combined setting, we use a distortion budget of $\epsilon = 0.01$,

---

[1]We did not see additional gains from using viewmakers on top of the full, well-optimized set of SimCLR augmentations.

[2]We do notice a smaller decline in contrast corruption accuracy, possibly due to interactions between changing pixel magnitudes and the $\ell_p$ constraint.

| ResNet-18, 100hr | Expert | | Ours ($\epsilon$) | | | ResNet-50, 960hr | Spec. | 0.05 |
|---|---|---|---|---|---|---|---|---|
| | Time | Spec. | 0.05 | 0.1 | | | | |
| LibriSpeech Sp. ID | **97.1** | 91.6 | 88.3 | 84.0 | | LibriSpeech Sp. ID | **95.9** | 90.0 |
| VoxCeleb1 Sp. ID | 5.7 | 7.8 | **12.1** | 9.1 | | VoxCeleb1 Sp. ID | 8.6 | **10.7** |
| AudioMNIST | 31.7 | 63.9 | **93.3** | 87.9 | | AudioMNIST | 80.2 | **88.0** |
| Google Commands | 27.1 | 31.9 | **47.4** | 41.6 | | Google Commands | 28.3 | **32.6** |
| Fluent Actions | 29.4 | 32.0 | **41.6** | 37.9 | | Fluent Actions | 30.5 | **42.5** |
| Fluent Objects | 37.1 | 40.3 | **47.6** | **47.6** | | Fluent Objects | 36.2 | **50.8** |
| Fluent Locations | 59.7 | 63.3 | 66.5 | **68.3** | | Fluent Locations | 62.0 | **68.9** |

Table 2: **Our learned views significantly outperform existing views for speech transfer tasks.** Linear evaluation accuracy for SimCLR models trained on LibriSpeech. Left: ResNet-18 + Librispeech 100 hour, Right: ResNet-50 + Librispeech 960hr. "Time" refers to view functions applied in the time domain (Kharitonov et al., 2020), while "Spec." refers to view functions applied directly to the spectrogram (Park et al., 2019). 0.05 and 0.1 denote viewmaker distortion bounds $\epsilon$.

which we find works better than $\epsilon = 0.05$, likely because combining the two augmentations at their full strength would make the learning task too difficult.

These results suggest that learned views are a promising avenue for improving robustness in self-supervised learning models.

## 5 SPEECH

Representation learning on speech data is an emerging and important research area, given the large amount of available unlabeled data and the increasing prevalence of speech-based human-computer interaction (Latif et al., 2020). However, compared to images, there is considerably less work on self-supervised learning and data augmentations for speech data. Thus, it is a compelling setting to investigate whether viewmaker augmentations are broadly applicable across modalities.

### 5.1 SELF-SUPERVISED LEARNING SETUP

We adapt the contrastive learning setup from SimCLR (Chen et al., 2020a). Training proceeds largely the same as for images, but the inputs are 2D log mel spectrograms. We consider both view functions applied in the time-domain before the STFT, including noise, reverb, pitch shifts, and changes in loudness (Kharitonov et al., 2020), as well as spectral views, which involve masking or noising different parts of the spectrogram (Park et al., 2019). To generate learned views, we pass the spectrogram as input to the viewmaker. We normalize the spectrogram to mean zero and variance one before passing it through the viewmaker, and do not clamp the resulting perturbed spectrogram. See the Appendix for more details. We train on the Librispeech dataset (Panayotov et al., 2015) for 200 epochs, and display some examples of learned views in the Appendix.

### 5.2 SPEECH CLASSIFICATION RESULTS

We evaluate on three speech classification datasets: Fluent Speech Commands (Lugosch et al., 2019), Google Speech Commands (Warden, 2018), and spoken digit classification (Becker et al., 2018), as well as speaker classification on VoxCeleb (Nagrani et al., 2017) and Librispeech (Panayotov et al., 2015), all using the linear evaluation protocol for 100 epochs. In Table 2, we report results with both the same distortion budget $\epsilon = 0.05$ as in the image domain, as well as a larger $\epsilon = 0.1$, for comparison. Both versions significantly outperform the preexisting waveform and spectral augmentations, with a +9 percentage point improvement on average for the ResNet-18 ($\epsilon = 0.05$) viewmaker model over the best expert views. The gains for real-world tasks such as command identification are compelling. One notable exception is the task of LibriSpeech speaker identification. Since LibriSpeech is the same dataset the model was pretrained on, and this effect is not replicated on VoxCeleb1, the other speaker classification dataset, we suspect the model may be picking up on dataset-specific artifacts (e.g. background noise, microphone type) which may make the speaker

| | Spectral | | Ours ($\epsilon$) | | | | |
|---|---|---|---|---|---|---|---|
| Dataset | With Noise | Without Noise | 0.02 | 0.05 | 0.2 | 0.5 | 2.0 |
| Pamap2 | 71.0 | 74.6 | 83.0 | 87.4 | 88.6 | **91.3** | 9.1 |

Table 3: **Our learned views significantly outperform existing views for activity recognition on wearable sensor data.** Our method learns superior representations across a large range of distortion budgets $\epsilon$, although budgets that are too strong prevent learning. Linear evaluation accuracy for ResNet18 models trained on Pamap2 with SimCLR. "Spectral" refers to view functions applied directly to the spectrogram (Park et al., 2019).

ID task artificially easier. An interesting possibility is that the worse performance of viewmaker views may result from the model being able to identify and ablate such spurious correlations in the spectrograms.

## 6 WEARABLE SENSOR DATA

To further validate that our method for learning views is useful across different modalities, we consider time-series data from wearable sensors. Wearable sensor data has a broad range of applications, including health care, entertainment, and education (Lara & Labrador, 2012). We specifically consider whether viewmaker views improve representation learning for the task of human activity recognition (HAR), for example identifying whether a user is jumping rope, running, or cycling.

### 6.1 SELF-SUPERVISED LEARNING SETUP

We consider the Pamap2 dataset (Reiss & Stricker, 2012), a dataset of 12 different activities performed by 9 participants. Each activity contains 52 different time series, including heart rate, accelerometer, gyroscope, and magnetometer data collected from sensors on the ankle, hand, and chest (all sampled at 100Hz, except heart rate, which is sampled at approximately 9Hz). We linearly interpolate missing data, then take random 10s windows from subject recordings, using the same train/validation/test splits as prior work (Moya Rueda et al., 2018). To create inputs for our model, we generate a multi-channel image composed of one 32x32 log spectrogram for each sensor time-series window. Unlike speech data, we do not use the mel scale when generating the spectrogram. We then normalize the training and validation datasets by subtracting the mean and then dividing by the standard deviation of the training dataset.

We train with both our learned views and the spectral views (Park et al., 2019) that were most successful in the speech domain (for multi-channel spectral masking, we apply the same randomly chosen mask to all channels). We also compare against a variant of these views with spectrogram noise removed, which we find improves this baseline's performance.

### 6.2 SENSOR-BASED ACTIVITY RECOGNITION RESULTS

We train a linear classifier on the frozen encoder representations for 50 epochs, reporting accuracy on the validation set. We sample 10k examples for each training epoch and 50k examples for validation. Our views significantly outperform spectral masking by 12.8 percentage points when using the same $\epsilon = 0.05$ as image and speech, and by 16.7 points when using a larger $\epsilon = 0.5$ (Table 3). We also find that a broad range of distortion budgets produces useful representations, although overly-aggressive budgets prevent learning (Table 3). These results provide further evidence that our method for learning views has broad applicability across different domains.

### 6.3 SEMI-SUPERVISED EXPERIMENTS

An especially important setting for self-supervised learning is domains where labeled data is scarce or costly to acquire. Here, we show that our method can enable strong performance when labels for only a single participant (Participant 1) out of seven are available. We compare simple supervised learning on Participant 1's labels against linear evaluation of our best pretrained model, which was

trained on unlabeled data from all 7 participants. The model architectures and training procedures are otherwise identical to the previous section. As Figure 4 shows, pretraining with our method on unlabeled data enables significant gains over pure supervised learning when data is scarce, and even slightly outperforms the hand-crafted views trained on all 7 participants (cf. Table 3).

| Dataset | Supervised Learning | | Pretrain (Ours) & Transfer | |
|---|---|---|---|---|
| | 1 Participant | 7 Participants | 1 Participant | 7 Participants |
| Pamap2 | 58.3 | 97.1 | 75.1 | 91.3 |

Table 4: **Our method enables superior results in a semi-supervised setting where labels for data from only one participant are available.** Validation accuracy for activity recognition on Pamap2. Supervised Learning refers to training a randomly initialized model on the labeled data until convergence. Pretrain & Transfer refers to training a linear classifier off of the best pretrained model above. 1 or 7 Participants refers to the number of participants comprising the training set.

## 7 CONCLUSION

We introduce a method for learning views for unsupervised learning, demonstrating its effectiveness through strong performance on image, speech, and wearable sensor modalities. Our novel generative model—viewmaker networks—enables us to efficiently learn views as *part of* the representation learning process, as opposed to relying on domain-specific knowledge or costly trial and error. There are many interesting avenues for future work. For example, while the $\ell_1$ constraint is simple by design, there may be other kinds of constraints that enable richer spaces of views and better performance. In addition, viewmaker networks may find use in supervised learning, for the purposes of data augmentation or improving robustness. Finally, it is interesting to consider what happens as the viewmaker networks increase in size: do we see performance gains or robustness-accuracy trade-offs (Raghunathan et al., 2019)? Ultimately, our work is a step towards more general self-supervised algorithms capable of pretraining on arbitrary data and domains.

ACKNOWLEDGEMENTS

We would like to thank Dan Yamins, Chengxu Zhuang, Shyamal Buch, Jesse Mu, Jared Davis, Aditi Raghunathan, Pranav Rajpurkar, Margalit Glasgow, and Jesse Michel for useful discussions and comments on drafts. AT is supported by an Open Phil AI Fellowship. MW is supported by the Stanford Interdisciplinary Graduate Fellowship as the Karr Family Fellow.

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
