# OpenReview forum: "Viewmaker Networks: Learning Views for Unsupervised Representation Learning"
_ICLR.cc/2021/Conference — ICLR 2021 Poster_

### Official Review · AnonReviewer1 · 2020-10-26
**Well motivated approach but experimental results are underwhelming on standard image datasets**

**Rating:** 6
**Confidence:** 3

**Review:**

Learning representations using self-supervision requires domain expertise to identify diverse transformations of the data samples that label preserving. This can be expensive and hard to obtain in many data modalities. The paper proposes to automate this by learning to generate transformations tailed to each modality and sample. Specifically, an adversarial strategy is applied to learning transformations that are close to the original view in the input space but hard to classify for the self-supervision encoder.

Pros:
+ The problem is well motivated and the approach is easy to follow.
+ Results are shown for multiple modalities with good performance including speech and time-series from wearable sensors.

Cons + Questions:
+ Adversarial Data Augmentations [1] approaches can be a good baseline to compare with in addition to "expert" views.
+ It will be helpful to elaborate on the claim - "model generates perturbations beyond training data."
+ N is not defined in Sec. 3.
+ Does constraining the norm of the perturbations to limit the possible transformations to the low-frequency range. Without encouraging some structure in perturbations, it may not be possible to obtain certain transformations such as localized perturbations. This could explain the lower performance against handcrafted transformations. Would the adversarial perturbation in the latent space help to produce more semantic image transformations.

[1] Zhang, Xinyu, et al. "Adversarial autoaugment." arXiv preprint arXiv:1912.11188 (2019).

---------
Thanks for the clarifications. I have raised my score.

I agree that the method is easier and more general than methods such as Adversarial Autoaugment. It will be interesting to see how the approach generalizes to larger/complex datasets without an expert specified family of transformations or without a good generative model.

---

> ### Author Response · Authors · 2020-11-17
> **Response to R1**
>
> We thank R1 for constructive and positive review, finding our work "well motivated" and "easy to follow," with "good performance" on multiple modalities.
>
> **R4 finds our method has "good performance including speech and time-series from wearable sensors" but that "experimental results are underwhelming on standard image datasets"** \
> Viewmaker performance on image transfer datasets matches those of expert augmentations (including cropping-and-resizing, many kinds of color, hue, and brightness transformations, and gaussian blurring). We believe it is significant that our method, with a simple \ell_1 bound, was able to match the performance of a complex and well-studied set of augmentations. Furthermore, the robustness gains are significant in the image domain when composed with the expert augmentations, closing 1/3 the gap to clean accuracy. Nevertheless, we believe future research may be able to develop viewmaker systems that significantly outperform expert augmentations even for visual domains.
>
> **"N is not defined in Sec. 3."** \
> Thank you for the catch! We'll note that it stands for the batch size in the next revision.
>
> **"Adversarial Data Augmentations [1] approaches can be a good baseline to compare with in addition to "expert" views."** \
> Thank you for this suggestion. We looked into Adversarial AutoAugment, but were unable to find code for the method online. Adversarial AutoAugment relies on REINFORCE to compute gradients with respect to discrete choices, which can be challenging to implement well and can suffer from poor variance or bias. The method also requires an expert-specified set of handcrafted augmentations and strength ranges. We find viewmakers to be a simpler and more general alternative, since they are fully differentiable and do not require a set of handcrafted augmentation families.
>
> **It will be helpful to elaborate on the claim - "model generates perturbations beyond training data."** \
> Thank you, we'll elaborate on this. By "beyond variation observed in the training data" we refer to the cited works which present models capturing naturally-occurring variation in the data (e.g. dimensions such as brightness/darkness or class variation). By contrast, viewmakers can generate complex kinds of perturbations not present in the training set, potentially improving generalization to new domains and creating a more challenging pretraining task.
>
> **"Does constraining the norm of the perturbations to limit the possible transformations to the low-frequency range. Without encouraging some structure in perturbations, it may not be possible to obtain certain transformations such as localized perturbations."** \
> We may be misunderstanding R1 here, and would be happy to clarify more in followup comments. As Figure 1 shows, the \ell_1 constraint allows for very localized perturbations in the input space, enabling both high- and low-frequency perturbations. Indeed, encouraging sparse perturbations is a major benefit of an \ell_1 constraint over an \ell_\inf constraint.
>
> **"Would the adversarial perturbation in the latent space help to produce more semantic image transformations."** \
> This is certainly an interesting direction for future work. One challenge is that this may necessitate a pretrained generative model so that the latent space already captures meaningful semantic dimensions.

---

### Official Review · AnonReviewer2 · 2020-10-27

**Rating:** 6
**Confidence:** 3

**Review:**

The paper proposes a method for automatic generation of data views for contrastive self-supervised learning of representations. The method consists of learning an adversarial perturbation model that aims to maximize the distance between the original image and its perturbed views in the space of learned representations. To avoid collapse to a completely information-destroying perturbation model, authors propose to limit the perturbation strength in terms of the $l_p$ norm of the added noise. Authors apply their method on various image, speech and wearable sensor datasets where the proposed approach provides an improvement over other methods.

Paper strengths:
1. The general idea of automating data augmentations for self-supervised learning seems very reasonable to me and adversarial training framework is a very viable option to implement these ideas.
2. The method does show an empirical improvement over SimCLR.

Paper weaknesses:
1. Recent and relevant work of Grill et al, 2015 "Bootstrap your own latent: A new approach to self-supervised Learning" is not even mentioned and not used as a baseline in the experimental comparison.
2. The particular form of additive perturbation considered in the paper seems somewhat limiting to me. For example, various rotations, contrast and saturation manipulations in the image domain are very difficult to reproduce within this framework and would result into a very high $l_p$ difference from the original image. It would be interesting to see if these transformations when added to the learned ones result into even better performance.
3.  While I do appreciate a quite broad selection of data domains in the experiments, I think it is important to also evaluate the proposed method on ImageNet since it is a major and established benchmark with enough complexity that makes good representations really necessary for a god performance.

Overall, I like the paper and would gladly increase my rating if authors could address the points above.

---

> ### Author Response · Authors · 2020-11-17
> **Response to R2**
>
> We thank R2 for the constructive and positive review.
>
> **"Recent and relevant work of Grill et al, 2015 "Bootstrap your own latent: A new approach to self-supervised Learning" is not even mentioned and not used as a baseline in the experimental comparison."** \
> Thanks for this catch—a more recent version of our paper mentions that our method is fully compatible with BYOL, in addition to other view-based methods like MoCo, Local Aggregation, etc. We'll be sure to include that in the next revision.
>
> **The particular form of additive perturbation considered in the paper seems somewhat limiting to me. For example, various rotations, contrast and saturation manipulations in the image domain are very difficult to reproduce within this framework and would result into a very high difference from the original image.** \
> It is certainly surprising that such a simple perturbation family enables such strong results. While the \ell_1 constraint performs well, we believe there is ample opportunity for developing other kinds of domain-agnostic constraints, whether through other kinds of metrics (e.g. Wasserstein distance) or other ways of bounding the complexity of the perturbation (e.g. applying the perturbations in frequency space, as we explore).
>
> **It would be interesting to see if these transformations when added to the learned ones result into even better performance.** \
> Great suggestion, thanks! We only had time to run transfer for a single Combined model (below), and did not see improvements in downstream performance. However, it is certainly possible that further gains could be realized with different perturbation budgets or constraints, higher-resolution inputs, or domains where augmentations are less developed.
>
> MNIST | 96.5 \
> traffic | 93.6 \
> FaMNIST | 88.8 \
> flower | 53.0 \
> aircraft | 28.9 \
> birds | 10.5
>
>
> **"While I do appreciate a quite broad selection of data domains in the experiments, I think it is important to also evaluate the proposed method on ImageNet"** \
> We also appreciate the importance of scale in both models and dataset size, and would like to see our method trained with larger models and more epochs on ImageNet as well as speech datasets. Unfortunately, we simply lack the resources needed to competitively train models at this scale. However, we hope the breadth of domains and transfer tasks we explore, in addition to the semi-supervised experiments in the wearable sensors domain, can highlight the benefit of our approach for the significant number of practitioners who have relatively modest computing resources.

---

> > ### Comment · AnonReviewer2 · 2020-11-25
> > **Still no comparison to BYOL**
> >
> > Thank you for your reply. Unfortunately, my concerns regarding the lack of side-by-side comparison with BYOL as well as discussing it remains. Authors did not update their manuscript so I can not assess how they envision potential compatibility with BYOL and since no experimental results were posted with this regard as well, I lack the grounds for revising my score.

---

### Official Review · AnonReviewer3 · 2020-10-28
**Interesting work for learning augmentation for contrastive learning**

**Rating:** 7
**Confidence:** 3

**Review:**

**[Summary]**
The authors proposed the Viewmaker, which learns to generate augmentation for contrastive learning. They show that the method achieves comparable results when applied for CIFAR-10, but significantly outperformed baseline augmentations in the speech domain and wearable sensor domain.

**[Reason for rating]**
I consider this work to be simple and effective. There might be other ways of learning augmentations for unsupervised or self-supervised learning, but the authors show one possible solution and demonstrate its effectiveness across image, speech, and wearable sensor domains. I have a few concerns as stated below, but I am leaning towards acceptance at this time.

**[Pros]**
- Propose a novel generative model that learns to augment inputs
- insightful analysis from Sec. 4.1.1 to 4.2
- Show the proposed method works for image, speech, and wearable sensors.
- The proposed method is well-motivated in the abstract., where expert-designed augmentations hinder the widespread adoption of self-supervised representation learning methods across domains and modalities.

**[Cons/Questions]**
- Viewmaker cannot make augmentations like cropping-and-rescaling (as already discussed by the authors). The authors already showed that the "Combined"(coming proposed Viewmaker and SimCLR augmentations) is robust against CIFAR-C corruptions in Table 4(a). I wonder how will the "Combined" performs for datasets in Table 1 (if the authors can kindly consider running this as additional experiments, it doesn't have to be all datasets).
- The authors claim that Viewmaker enables significant robustness against CIFAR-C corruptions, but we (readers) can only see that the Viewmakes performs the worst in Table 4(a). Perhaps this claim should be validated by comparing it with a baseline where traditional data augmentation is used.

**[Minor comments]**
- It's not clear to me what's the Expt in Table 1 at first glance. Perhaps indicate that it stands for Expert view in the table caption.
- Table 2 was not referred to in Sec. 5.2.
- Across the tables, Ours, Learned, and Viewmaker are used to refer to the proposed method. Can we make it consistent?

---
**[After rebuttal]**

Thanks for the clarifications. My score remains the same after reading through all the other reviews and the replies from the author. During the rebuttal, the authors attempt to answer many of the questions raised by the reviewers. While some of the replies are less satisfying, I still find this work to be worth to publish. I would encourage the authors incorporate the changes in the revision if the paper was accepted.

---

> ### Author Response · Authors · 2020-11-17
> **Response to R3**
>
> We thank R3 for the useful review, finding our method "well motivated," "simple and effective."
>
> **"Viewmaker cannot make augmentations like cropping-and-rescaling (as already discussed by the authors)"** \
> We'll add some more discussion about this: In some sense there is a tradeoff between proposing a domain-agnostic method for pretraining and introducing useful domain-specific transformation (e.g. cropping-and-resizing). For example, while horizontal flipping may be useful for CIFAR, we may not wish to use it when training on chest x-rays where the heart will (hopefully!) always appear on the same side of the image. Thus, while \ell_1 perturbations are a compelling domain-agnostic perturbation family which performs surprisingly well across a range of tasks, we view it as a foundation upon which other domain-specific augmentations may be added if they exist.
>
> **"I wonder how will the "Combined" performs for datasets in Table 1 (if the authors can kindly consider running this as additional experiments, it doesn't have to be all datasets)."** \
> This is a great suggestion, thanks! We only had time to run transfer for a single Combined model (below), and did not see improvements in downstream performance. However, it is certainly possible that further gains could be realized with different perturbation budgets or constraints, higher-resolution inputs, or domains where augmentations are less developed.
>
> MNIST | 96.5 \
> traffic | 93.6 \
> FaMNIST | 88.8  \
> flower | 53.0  \
> aircraft | 28.9  \
> birds | 10.5
>
>
> **"The authors claim that Viewmaker enables significant robustness against CIFAR-C corruptions, but we (readers) can only see that the Viewmakes performs the worst in Table 4(a)."** \
> Thanks, we'll clarify this in the next version. It is challenging to find an effective baseline as CIFAR-C corruptions overlap with common augmentations used to train vision models, but we will find a better wording here.
>
> **[Minor comments]** \
> Thank you for these! We'll include them in the next version.

---

### Official Review · AnonReviewer4 · 2020-10-28
**A well-written paper on a new generative model to generate data for contrastive learning, with some deficiency in experiments**

**Rating:** 7
**Confidence:** 4

**Review:**

The paper presents a generative model to automatically generate data that is needed for contrastive learning, with a focus on the SimCLR framework, while the method itself is general. Experiments were conducted across multiple modalities, including image, speech and wearable sensor data. The results demonstrate the effectiveness of the proposed model as compared to data augmentation methods relying on human domain knowledge.

The paper is generally well-written with a clear introduction to the state-of-the-art, a strong motivation to fill in a gap in the literature for automating data augmentation for self-supervised representation learning, and a concise description of the proposed method. It addresses a relevant and timely problem, and the method is technically sound. Evaluation is thorough including experiments across modalities and data sets, in addition to an ablation study. The effectiveness of the proposed methods has been demonstrated.

There are several points that are yet to be clarified or discussed as detailed below.

In the abstract, it is said that the proposed method significantly outperforms baseline augmentations in speech (+9% absolute). As a number of experiments for different data sets with different hyper-parameters were conducted, it is unclear where the 9% absolute improvement comes from and this is not mentioned in the main body of the paper either.

The viewmaker network aims at generating complex perturbations to the input data. What about generating perturbed data directly in a way similar to adversarial example generation?

Experiments were conducted on a number of speech data sets for different tasks, which is nice. On the other hand, the performance in terms of accuracy seems to be fairly low for several tasks as compared to the performance obtained by state-of-the-art methods in the literature.
- For example, on the Google Commands data set, accuracies of between 27.1% and 47.4% obtained in this paper are far below the baseline ones in the original dataset paper, which are between 82.7%-89.7% (depending on the settings), and the numbers are higher in latest works in the literature.
- In another example, for the VoxCeleb data set, the accuracies for speaker identification are between 5.7%-12.1% in this paper, while the Top-1 accuracies are between 49.0%-80.5% for the baseline methods in the original dataset paper. Assuming Top-1 accuracy is used as the metric in this paper; otherwise Top-5 accuracy is even higher, which is good to specify in the paper as well.
- Even though there might be differences in experimental settings and a linear classifier is used here, the gap between the results here and in the literature is too significant to make a solid conclusion based on the results, as the gain can potentially disappear when a stronger backend classifier is applied. Good to discuss this.
- In general, the experimental settings for the speech experiments are unclear even after reading the supplementary material.

No statistical significant test or 95% confidence intervals (together with accuracy) or the like are presented for any experiments.

The reproducibility of the work could be enhanced with providing more details about the experimental settings.

---

> ### Author Response · Authors · 2020-11-17
> **Response to R4**
>
> We thank R4 for the positive and helpful review!
>
> **Origin of +9% number for speech** \
> Thanks, we'll clarify this in the paper—this is the average of the absolute difference in accuracy for the Resnet-18 model with \eps=0.05 on speech transfer tasks.
>
> **"What about generating perturbed data directly in a way similar to adversarial example generation?"** \
> This is an interesting idea we considered, though there are a few challenges for straightforward approaches less promising. First, the view generator should be able to generate multiple diverse views from a single image, while most processes for generating adversarial examples are approximately deterministic (modulo small differences due to random seeds). Second, multiple iterations of e.g. PGD are expensive, while our method can amortize that cost and even use a smaller, cheaper model to do so.
>
> **"Even though there might be differences in experimental settings and a linear classifier is used here, the gap between the results here and in the literature is too significant to make a solid conclusion based on the results, as the gain can potentially disappear when a stronger backend classifier is applied. Good to discuss this."** \
> We'll be sure to characterize this clearly in the next revision, that we are mainly comparing to an equivalent SimCLR model on linear evaluation, and not larger, supervised SOTA models trained end-to-end on the transfer task, which would be expected to have much higher accuracy.
>
> **"Top-5 accuracy [for VoxCeleb]… is good to specify in the paper as well."** \
> Great suggestion, we have these numbers here and will include them in the next revision:
>
> resnet18, 100hr \
> expert time |  top1: 0.05774, | top5: 0.13216 \
> expert spectral | top1: 0.07666 | top5: 0.16076 \
> viewmaker 0.05  | top1: 0.13122 | top5: 0.24987 \
> viewmaker 0.1    | top1: 0.09753 | top5: 0.20434
>
> resnet50, 960hr \
> expert spectral   | top1: 0.08618 | top5: 0.16882 \
> viewmaker 0.05  | top1: 0.10751 | top5: 0.21923
>
> **"In general, the experimental settings for the speech experiments are unclear even after reading the supplementary material."** \
> Thank you for letting us know. We will be releasing our code with all the experimental settings, but if there are specific speech details you find lacking or confusing in the text, we'd be happy to remedy.
>
> **"No statistical significant test or 95% confidence intervals (together with accuracy) or the like are presented for any experiments."** \
> Thank you for this suggestion. We have conducted additional pretraining and transfer runs for a majority of image transfer tasks for the SimCLR CIFAR-10 models (both the expert and viewmaker views) and added standard deviations. These demonstrate the stability of our approach. We will add these for the remaining tasks in the next revision.
>
> **Dataset**  |   Viewmaker | Std. Dev | Expert | Std. Dev \
> **aircraft**   |   32.0              | 0.7       |    32.0    |  0.6 \
> **birds**  | 8.7                | 0.3          |  10.9     | 0.3 \
> **dtd**            | 27.8              | 0.9         |   30.4    |  1.1 \
> **FaMNIST**  | 91.0              | 0.4         |   88.5    |  0.2 \
> **MNIST**        | 98.8              | 0.1         |   97.1    |  0.0 \
> **traffic**       | 94.8              | 1.0          |   96.7    | 0.3 \
> **flower**      | 50.6              | 2.6           |  53.2   |  0.4

---

### Public Comment · ~Yonglong_Tian1 · 2020-11-10
**Nice work, Congrats! Potential connection with related work**

Hi Authors,

It's nice to see the ideas of learning views for contrastive learning.

But I think the adversarial approach of learning views is closely related to the section 4 of the following paper:

what makes for good views for contrastive learning? https://papers.nips.cc/paper/2020/file/4c2e5eaae9152079b9e95845750bb9ab-Paper.pdf

In this paper, two approaches for view learning that also uses minimax (or adversarial) formulation is provided: a purely unsupervised view learning approach and a semi-supervised learning approach that is aware of the downstream task.

Currently I did not find a section that describes the similarity and dissimilarity between yours approach and the above paper (ours). I wonder how you think. Thanks!

---

> ### Author Response · Authors · 2020-11-17
> **Response**
>
> Thanks for your kind comments and for sharing your paper! We do cite this work in our paper, and are happy to make clear that you also explore minimax setups.
>
> Our understanding from reading section 4 was that the proposed color-space learning method required knowledge of the specific transfer task to be successful, as the unsupervised baseline performed comparably or worse than the raw inputs. In contrast, our focus is unsupervised representation learning, and viewmakers enable successful unsupervised learning of views without specializing the representations for a particular transfer task, so that a single model can transfer to a wide range of tasks. Another difference is that viewmakers don't require domain-specific assumptions (e.g. color-space or multichannel inputs), enabling use across different modalities.

---

### Decision · Program_Chairs · 2021-01-07
**Final Decision**

**Decision:**

Accept (Poster)

**Comment:**

New generative model to come up with data that is needed when doing contrastive learning. Like the fact that multiple modalities were considered and evaluated. The Viewmaker methods appears to do well on CIFAR-19 and outperforms baselines on speed and wearable domains. The reviewers praise the method for being simple, well-described and well-motivated. The main drawbacks stem from the fact that viewmaker cannot make certain types of image-specific augmentations (crop & rescale, as an example), but it's fair that the authors argue that their method is more domain-agnostic; and one can indeed add more domain-specific stuff if needed.

All in all, this seems like a solid paper with an easy to implement idea that is quite general and that has been shown to work in a variety of settings. It definitely belongs at ICLR.